# The 2 Sigma Genus Concept in mammalogy: Lessons from *Lasiurus*

Amy B. Baird[1]*, Janet K. Braun[2], Mark D. Engstrom[3], Burton K. Lim[3],
Michael A. Mares[2], Luis A. Ruedas[4], John C. Patton[5], John W. Bickham[6]

1 Department of Natural Sciences, University of Houston - Downtown, Houston, Texas, United States of America, 2 Sam Noble Oklahoma Museum of Natural History, University of Oklahoma, Norman, Oklahoma, United States of America, 3 Royal Ontario Museum, Toronto, Ontario, Canada, 4 Department of Biology and Museum of Vertebrate Biology, Portland State University, Portland, Oregon, United States of America, 5 5000 Quartz Lane, Midland, Texas, United States of America, 6 Department of Ecology and Conservation Biology, Texas A&M University, College Station, Texas, United States of America

* bairda@uhd.edu

## Abstract

Species concepts are well established and apply across diverse groups of organisms; however, there is no consensus on what defines higher taxonomic groups. The genus rank is important to taxonomists because it comprises part of the scientific name of an organism. A consistent and biologically meaningful method for determining generic status is needed for taxonomic stability and utility. Lasiurine bats are a group for which there is disagreement on how many genera to recognize. Some authors argue for splitting this group into three genera based on morphology, genetic divergence, and time of divergence; others argue that a single genus should be maintained. Here, we use lasiurines to explore generic-level taxonomy and how it is applied. Genetic divergence levels are compared among sister genera and within genera of vespertilionine bats using Cytochrome *b* (*Cytb*) sequences. We used *Cytb* because it is the most sequenced mitochondrial gene in mammals, but other genes might be more appropriate for a different taxon. Future methods will eventually use complete mitogenomes and genomes. We conclude that lasiurine bats are most appropriately divided into three genera to maintain taxonomic consistency within their subfamily. Since Linnaeus, the quarter millennium of progress in the science of mammalogy has provided a binomial nomenclatural basis from which can be extracted an acceptable range of genetic diversity upon which to establish generic level taxonomy. We offer a biologically meaningful operational definition of the genus, which we call the 2 Sigma Genus Concept, based on genetic divergence between a genus and its sister genus or lineage and compared to the divergence between sister pairs of established genera in the same higher taxonomic category. Our method is phylogenetic; sister genera are based on the best phylogeny for the higher taxonomic category. Genera must be monophyletic and differ from their closest relatives by not more than two standard deviations above or below the mean value of genetic

**Data availability statement:** All relevant data are within the paper and its Supporting information files.

**Funding:** The author(s) received no specific funding for this work.

**Competing interests:** The authors have declared that no competing interests exist.

distance for the larger taxonomic group in which they are contained. There should be genetic-based characters (e.g., morphology, protein structure, behavior) that are diagnostic for each genus. Our method is novel in that it uses the statistical distribution of sister divergences within a higher category to guide the allocation of generic status to monophyletic lineages. Within Vespertilioninae, the mean genetic distance between sister genera is $20.68\% \pm 3.89\%$ (K2P) for the *Cytb* gene. Therefore, a proposed new genus should have >12.90% genetic distance to its sister genus. Genera that are >2 standard deviations above the mean (>28.46%) are candidates to be recognized as a higher category such as tribe or subfamily. The sister-genus divergence for the monophyletic lineage that includes all lasiurine bats is $31.14\% \pm 1.64\%$ which qualifies it as a higher category (i.e., tribe), and the sister-genus divergences of the three monophyletic lineages within Lasiurini (i.e., red, yellow and hoary bats) are 21.77% and 22.91% which qualifies them for genera. We show the applicability of the method beyond Vespertilioninae by providing a case study where we apply it in a distantly related subfamily of bats.

## Introduction

The species category [1] generally is accepted to be the only objective taxonomic rank and has, or potentially has, biological meaning because it is associated with reproductive isolation. This has led to the development of a multitude of species concepts, including the biological species concept, the most widely used framework, which has reproductive isolation as its central tenet [2–4] or the evolutionary species concept which focuses on recognition of distinct evolutionary lineages [5–7], among others. In contrast, higher taxonomic categories often are considered subjective constructs, and thus unnatural. The restriction of higher taxonomic ranks to monophyletic evolutionary lineages is consistent with the inherent hierarchical structure of diversification through lineage splitting in biological systems, and provides a framework, with the potential to objectively scale taxonomy. However, there is no general agreement as to thresholds for delineation between higher ranks. Genetic distance to estimate the age of lineages and the inability to produce intergeneric hybrids are two approaches that have been proposed to approximate thresholds, and while both have potential for success, they have not been uniformly applied [8,9]. Historically, higher taxa are recognized by morphological distinction and more recently by genetic divergence. Today, it is universally accepted that higher taxa must correspond to monophyletic lineages that are morphologically and genetically distinct. Among higher taxonomic categories, the category Genus is uniquely important because it constitutes part of the scientific names of the contained species. Moreover, Helgen et al. (p. 270) [10] recommended that "*Generic boundaries should effectively indicate evolutionary relationships not only for taxonomic consistency, but also because the genus is commonly employed as the level of analysis in phylogenetic, paleontological, macroevolutionary, and other comparisons.*" Notwithstanding the importance

of this taxonomic category, there are no generally accepted guidelines for the recognition of genera. But morphologically and genetically distinct monophyletic lineages are required as for all higher categories. Beyond this, there are no genus concepts that parallel the multitude of species concepts that have been proposed in the systematics literature.

New genera typically are erected when the species constituting a genus are found not to be reciprocally monophyletic. Often this entails a molecular phylogenetic study that reveals an unexpected paraphyly. A new genus-level taxonomy then is necessary for all the constituent species to belong to monophyletic taxa. An example of this is the ground squirrel genus *Spermophilus* Cuvier, 1825 [11], which prior to the taxonomic revision by Helgen et al. [10] had an extensive Holarctic distribution and was comprised of 41 species and 6 subgenera. Molecular phylogenetic studies [12,13] using Cytochrome b (*Cytb*) showed that the genera *Cynomys* Rafinesque, 1817 [14], *Marmota* Blumenbach, 1779 [15] and *Ammospermophilus* Merriam, 1892 [16] were nested within *Spermophilus*, rendering the latter genus paraphyletic. To resolve this taxonomic issue, Helgen et al. [10] undertook a generic revision of *Spermophilus* based on the following principles: "*Delineation of revised generic boundaries in this case ideally requires not only a well-resolved phylogeny, but also an overview of available generic names and their type species, critical considerations of generic definitions and content, and revised morphological diagnoses of recognized genera*" ( [10]: 271). Helgen et al. [10] recognized eight genera within what was formerly *Spermophilus*. These genera are monophyletic and morphologically distinct. This radically different taxonomy has been accepted with little or no dissent.

In some cases, however, monophyletic genera are proposed to be subdivided into multiple genera for reasons other than phylogenetic ones. For example, the monophyletic lizard genus *Anolis* Daudin, 1802 [17] is comprised of nearly 400 species of mainly Neotropical lizards. Due to the large number of species, Nicholson et al. [18] proposed a markedly changed taxonomy that subdivided *Anolis* into eight monophyletic genera. However, this proposal was not widely accepted, and *Anolis* remains a single genus with multiple named clades that do not correspond to Linnaean categories [19]. Thus, one solution to the recognition of an overly complex taxon appears to be to dispense with the Linnaean system of nomenclature. Herpetologists seem happy with this, but we agree with Teta (p. 209) [20] that there is advantage to the Linnaean system in which taxonomic names are "*governed by the rules of the International Commission on Zoological Nomenclature (ICZN) [21], having its usage constrained (and stability promoted) by typification and priority (cf. Voss et al. [22])*" ( [20]: 209). Teta was referring specifically to the use of subgenera, but the statement is true of all levels of the taxonomic hierarchy.

There are other examples where high diversity within a monophyletic genus has necessitated the subdivision of a genus into multiple genera; genera that are comprised of far fewer species than *Anolis*. In these cases, the genus contains multiple monophyletic lineages judged to be morphologically and genetically as diverse as other recognized genera within the same family. In such cases, the original genus stands out as being highly diverse, but more importantly its subdivision results in a more natural taxonomy that better represents the pattern of generic and morphological diversity within the family or other higher taxon in which it is contained. Examples in mammalian taxonomy include the subdivision of the chipmunk genus *Tamias* Illiger, 1811 [23] into three genera, recognizing in addition *Eutamias* Touessart, 1880 [24] and *Neotamias* Howell, 1929 [25,26]. Despite the original *Tamias* being monophyletic, recognition of three genera was based on morphological distinction among the three, and genetic divergence among the three, being equivalent to related sciurid rodent genera. An argument based on the estimated ages of the lineages also was made. An important point here is that the newly recognized genera should have genetic differentiation (which approximates similar ages of divergence) among each other on par with those among other recognized genera within the higher taxon (typically the family) in which they all are included.

Another example of a diverse monophyletic genus that was partitioned for the same reasons as *Tamias* is the tree bat genus *Lasiurus* Gray, 1831 [27]. Baird et al. [28] undertook a molecular systematic revision of the primarily New World vespertilionid tribe Lasiurini Tate, 1942 [29], which consisted prior to their study of a single genus, *Lasiurus*. Baird et al. [28] demonstrated that *Lasiurus* contained three monophyletic lineages of species colloquially referred to as red bats,

yellow bats, and hoary bats, due to their distinct coloration and ease of identification based on that character (Fig. 1). The red bats are reddish in color and relatively small in size, yellow bats are yellow or yellowish brown in color, variable in size, and hoary bats are large, with the most basal species having a red bat phenotype and the remaining three species possessing dark but frosty pelage (Fig 1). Their molecular analyses included DNA sequence data from three mitochondrial protein-coding genes (*ND1*, *ND2*, and *Cytb*) and one nuclear Y-chromosome gene (*DBY*). Phylogenetic analyses of individual loci, as well as a species tree based on all four loci congruently resolved relationships among the three monophyletic lineages and provided genetic distance estimates suggestive of generic distinction. The red bat lineage remained *Lasiurus*, the hoary bat lineage became *Aeorestes* Fitzinger, 1870 [30], and the yellow bat lineage became *Dasypterus* W. Peters, 1870 [31].

Several papers criticized the new generic concept of Baird et al. [28], including Ziegler et al. [32], Novaes et al. [33], Teta [20], and most recently a non-peer-reviewed commentary by Francis et al. [34]. Those authors did not dispute the genetic differentiation of the three monophyletic lineages but questioned the need to erect new genera, instead recommending that the new taxa be recognized as subgenera. One reason given in all these critiques was the opinion that Baird et al.'s [28,35] taxonomic arrangement went against the ICZN's principle of nomenclatural stability. The data they gave to support this were literature searches —showing—not unexpectedly—that Google Scholar searches based on the old name for the hoary bat (*Lasiurus cinereus*) return more hits than the new name (*Aeorestes cinereus*). The series of

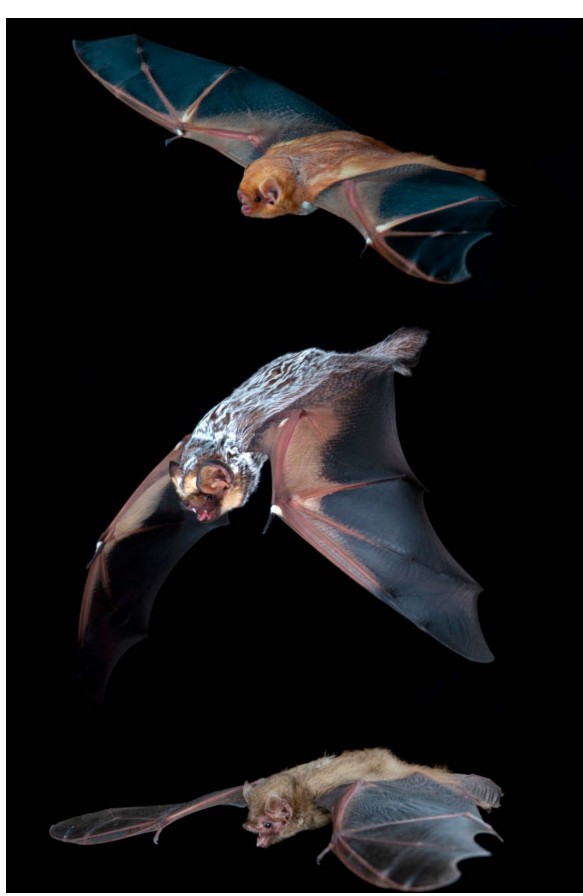

**Fig 1. Pictures of (top to bottom) *Lasiurus borealis*, *Aeorestes cinereus*, and *Dasypterus ega* depicting morphological characters typical of each genus.** Reprinted under a CC BY license, with permission from Brock Fenton, original copyright 2024.

opinion papers prompted responses by Baird et al. [35] who provided additional sequences from the mitochondrial COI and nuclear CMA1 and Rag2 loci, and Baird et al. [36], who calculated divergence times of the lasiurine lineages and compared them to other vespertilionid genera using published data from Amador et al. [37].

The most recent critique is the non-peer-reviewed commentary by Francis et al. [34]. The authors reviewed the above-listed exchange of views about one genus versus three for lasiurine bats with the purpose of recommending what taxonomy to use for the taxonomically influential websites Mammal Diversity Database [38] and Batnames.org [39]. Francis et al. [34] recommended "*retaining the genus* Lasiurus *for all species within the tribe Lasiurini, but recognizing three subgenera…*". With this recommendation, and a reading of the other opinion papers cited above, it should be clear that all authors accept that Baird et al. [28,35,36] have correctly identified three higher-level taxa within the tribe Lasiurini and that the only argument is the taxonomic level at which these newly erected taxa should be recognized, i.e., subgenera [20,32–34] versus genera [28,35–37,40–50]. To support their recommendation of a single genus of lasiurine bats, Francis et al. [34] offered four points: 1) "*the genus* Lasiurus *thus defined is monophyletic*"; 2) "*the use of subgenera is appropriate to recognize groupings within the genus*"; 3) "*there are no objective criteria for suggesting that a particular timing or degree of divergence merits elevating these subgenera to genera, and in any case there is considerable uncertainty in estimated divergence times of these groups and other genera in the family Vespertilionidae*"; and 4) "*finally that retention of the genus* Lasiurus *for all members of the tribe Lasiurini meets the ICZN [*21*] goal of promoting maximum stability in nomenclature*." We disagree with points 2, 3 and 4 and herein we examine whether *Lasiurus sensu lato* (referring to a single genus, *Lasiurus,* containing all species of lasiurine bats) is similar to other vespertilionine genera in terms of genetic distance between sister genera of vespertilionine bats. We will put the genetic distance of *Lasiurus sensu lato* in context with other genera in the subfamily Vespertilioninae and examine patterns of intra- and intergeneric distances to formulate a framework for defining a genus. We will test this on stenodermatine bats to determine if the use of genetic-based metrics holds promise of being the long-sought objective criterion mentioned in point 3 of Francis et al. [34].

Controversy and disagreement are not unusual in the science of taxonomy. The back-and-forth debate regarding the status of *Lasiurus* has caused us to formalize our concept of the category Genus. Herein, we describe our reasoning and methods in what we call the 2 Sigma Genus Concept.

## Materials and methods

### Sequence selection

We examined published *Cytb* sequences of bats in the subfamily Vespertilioninae to assess levels of divergence between sister genera and within polytypic genera. *Cytb* was chosen because it is frequently used in genus- and species-level phylogenetic studies of mammals [51,52], so a high percentage of vespertilionid species are represented on GenBank with this gene. Ideally, these analyses would be conducted with whole genomes and mitogenomes, but those data do not exist at this time across the necessary range of species or genera. Nonetheless, whole genome sequences eventually will be readily available and inexpensive [53,54]. We restricted most of our analyses to comparisons within and among genera within the highly diverse subfamily Vespertilioninae (within which there are 316 recognized species; [39]). In taxa less diverse than Vespertilioninae, a family-wide comparison may need to be made. We also provided a case study on stenodermatine bats to illustrate the broader applicability of our method.

### Sister genera analysis

To determine relationships within the subfamily Vespertilioninae and analyze divergence among sister genera, we followed the phylogeny published by Amador et al. ( [37]; their supplemental figure S6 depicting detailed relationships of Vespertilioninae, summarized in our Fig 2). We further subdivided *Eptesicus sensu lato* into *Eptesicus* Rafinesque, 1820 [56]*, Neoeptesicus* Cláudio, Novaes, Gardner, Nogueira, Wilson, Maldonado Oliveira, & Moratelli, 2023 [55], and *Cnephaeus* Kaup, 1829 [57] following Cláudio et al. [55]. We followed the taxonomy of Monadjem et al. [58] in their revision

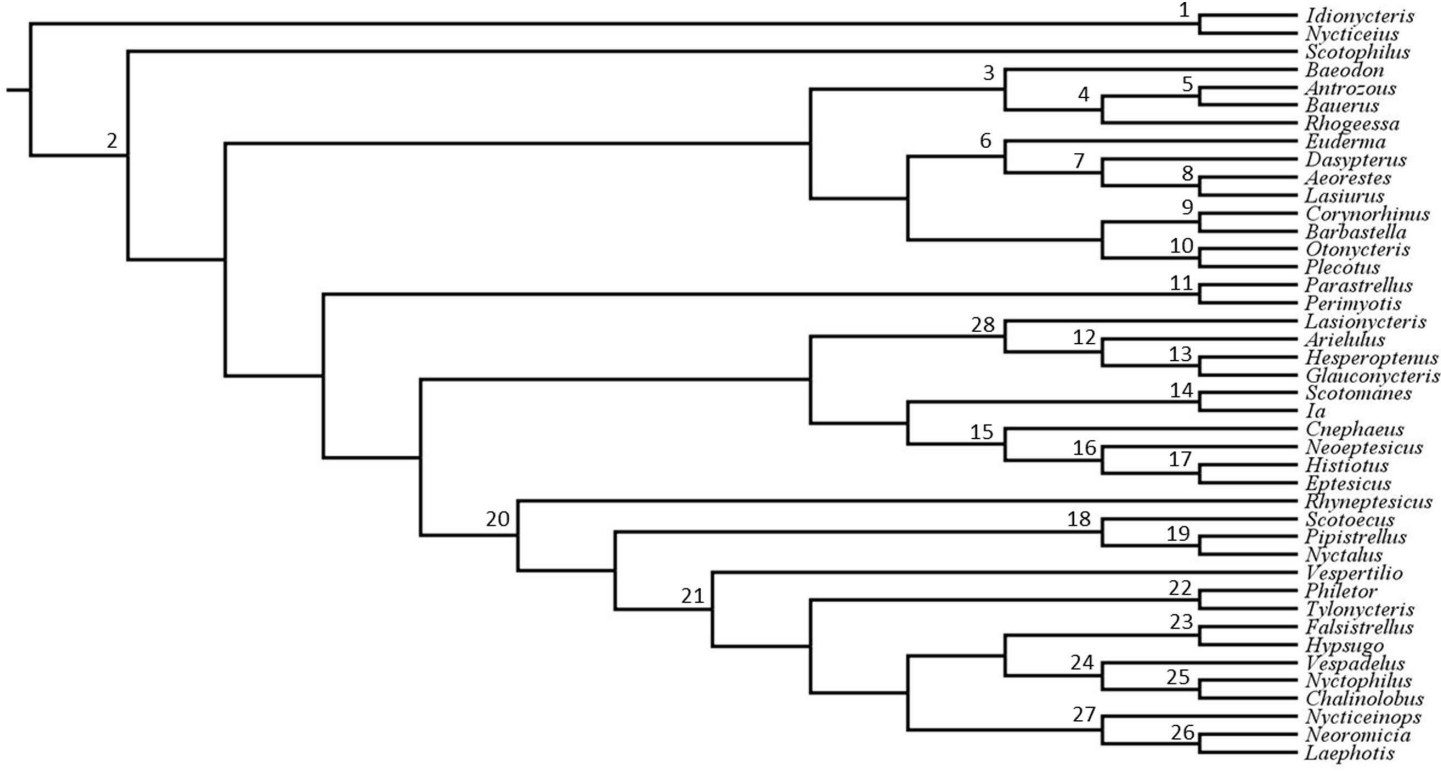

**Fig 2. Phylogeny of bats in the subfamily Vespertilioninae.** This phylogeny is derived from Amador et al. (24; their supplementary Fig. S6) with the exception of the subdivision of *Eptesicus sensu lato* into three genera following Cláudio et al. [55]. Numbers above nodes correspond to sister genera analyses shown in Fig 3.

of *Neoromicia* Roberts, 1926 [59]. These authors recognized the genera *Afronycteris* Monadjem, Patterson, & Demos, 2020 [58] and *Pseudoromicia* Monadjem, Patterson, & Demos, 2020 [58]. Because the relationship of the clade *Afronycteris*+*Pseudoromicia* is not well resolved with respect to *Neoromicia* and *Laephotis* Thomas 1901 [60], we did not use *Afronycteris* and *Pseudoromicia* in our analysis of sister genera. We thus limited the species within *Neoromicia* to those included by Monadjem et al. [58].

For all comparisons described below, we utilized one *Cytb* sequence per species (Supplementary S1 Table). Only when a *Cytb* sequence was unavailable for a given species was it eliminated from our analysis. For all pairs of genera for which there is only one sister taxon (i.e., each genus is sister to another genus rather than to a lineage of multiple genera), we averaged the distance of each available species in Genus A to each available species in Genus B and calculated the standard deviation of those distances.

Some genera were sister to a monophyletic lineage of multiple genera. A clade with the relationship (A, (B,C)) would be an example of such a sister genera relationship for Genus A. In this example, we would estimate the genetic distance between each species in the sister genus (A) to each species within the genera contained within the monophyletic lineage (B and C) and average the distances. This methodology should provide a conservative estimate of average genetic divergence, given that the outlying genus is likely to be relatively distantly related to the ingroup genera, and we are making multiple comparisons of that genus to the ingroup taxa. Thus, if the lasiurine taxa fall within two standard deviations of the mean sister divergence calculated this way, they could be considered quite distinct. The numbered clades in Fig 2 correspond to the common ancestor of the sampled sister genera. For example, node 3 corresponds to the common ancestor

of *Baeodon* Miller, 1906 [61] and its sister clade containing *Rhogeessa* H. Allen, 1866 [62], *Antrozous* H. Allen, 1862 [63], and *Bauerus* Van Gelder, 1959 [64].

Our top priority for sequence selection was length (full *Cytb* sequence was preferred). We aligned the sequences using Geneious v. 9.1.8 [65]. We used MEGA 11 [66] to calculate K2P distances among taxa and to find the most appropriate model of evolution for the data.

### Within genus analysis

To analyze divergences within genera, we used batnames.org [39] and Mammal Diversity Database [38] to find all genera in the subfamily Vespertilioninae that contained more than one species. The exceptions to the taxonomy used by both sources follows the recognition of three genera of lasiurines (*Aeorestes, Dasypterus, Lasiurus*; Baird et al. [28]) as well as the splitting of *Eptesicus sensu lato* into *Eptesicus, Neoeptesicus*, and *Cnephaeus* following Cláudio et al. [55]. Additionally, we followed the taxonomy of Monadjem et al. [58] in their revision of *Neoromicia.* We selected one *Cytb* sequence per species (again with a preference toward the longest sequence) for each of the genera. If only one species from a genus with a *Cytb* sequence could be found, that genus was eliminated from this analysis. Divergences within *Lasiurus sensu lato* and within *Eptesicus sensu lato* also were calculated along with the contained newly erected genera within them as described above. A list of all species and GenBank accession numbers used can be found in Supplementary S1 Table. All sequences for each genus were aligned and percent K2P sequence divergence was calculated using Geneious and MEGA 11, respectively, as described above. The average and standard deviation of K2P genetic distance within each genus was calculated based on averaging the value for each species pair divergence.

### Case study – Stenodermatinae

In a case study to compare how our method performs in another group of bats, we downloaded *Cytb* sequences from GenBank corresponding to bats in the subfamily Stenodermatinae (Chiroptera: Phyllostomidae; Supplementary S1 Table). The species tree derived from two mtDNA genes and one nuclear gene presented by Garbino and Tavares [67] was used to determine sister genera relationships and derive distances among sister genera, as described above for lasiurine bats (summarized in Supplementary S5 Fig.). The phylogeny of Garbino and Tavares [67] did not include the genus *Koopmania* Owen, 1991 [68], and previous studies [69–71] had conflicting results as to whether *Koopmania* is sister to *Artibeus* Leach, 1821 [72] or *Dermanura* Gervais, 1856 [73], so we examined both of those possible relationships. One of these arrangements, *Koopmania* as the sister taxon to *Dermanura*, is reflected in Supplementary S5 Fig. *Enchisthenes* and *Pygoderma* also were not included in Garbino and Tavares [67] and were eliminated from our analysis because their phylogenetic position has varied considerably in other studies [74].

As above, one *Cytb* sequence per species was used in the analysis. A matrix of K2P distances among each species was generated by Mega 11 [66]. Each species of one genus was compared to each species of its sister genus (or lineage), and the average and standard deviation of these distances were computed.

For both the lasiurine and stenodermatine sister genus distance datasets, we used a Shapiro-Wilk Test to determine whether the datasets were normally distributed [75].

## Results

### Sister genus analysis

The available data resulted in 29 sister genus comparisons among vespertilionine bats (Fig 3; Supplementary Data S2 Table). MEGA 11 identified GTR+Γ+I as the most appropriate model of evolution for the genus alignment. However, we report K2P distances here for consistency with other studies centered on genetic divergence of mammals, e.g., [51,52]. The values of distances between sister genera ranged from 13.21% ± 5.6% (*Neoeptesicus* – multiple) to 31.14% ± 1.64%

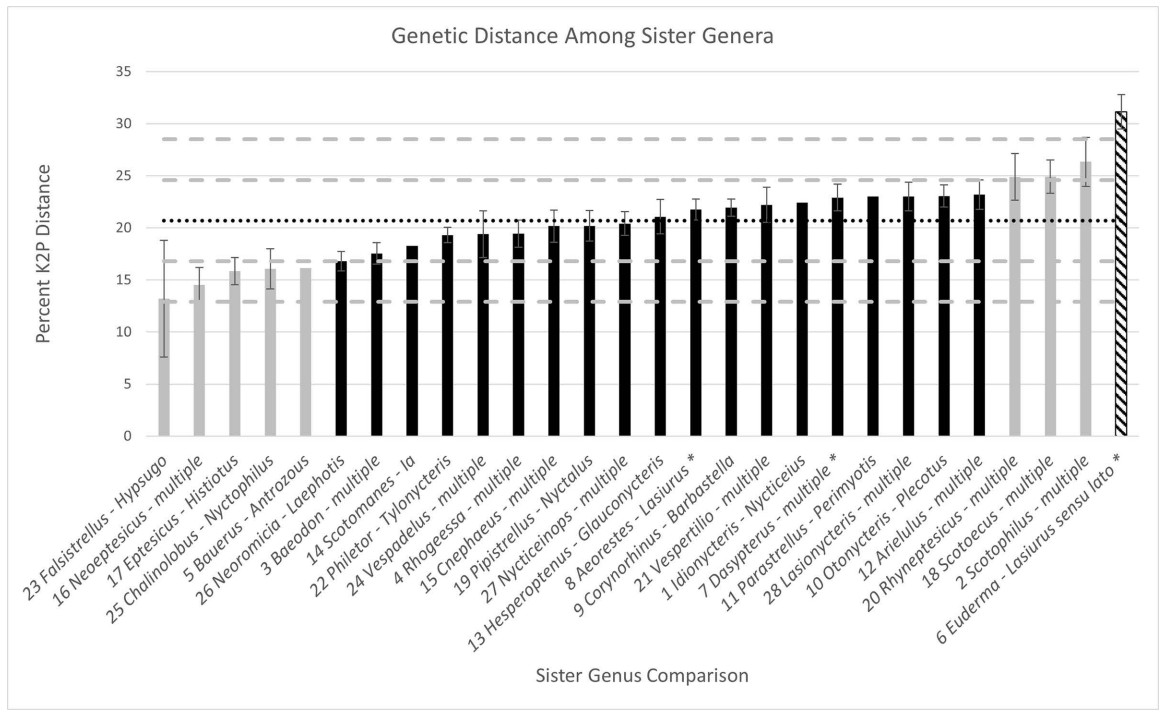

**Fig 3. Genetic distance among sister genera of vespertilionine bats.** Mean value (20.68%) is depicted by the horizontal black dotted line. Each standard deviation above and below the mean is depicted by a gray dashed line. Black bars represent divergence values within one standard deviation of the mean (16.79% − 24.57%). Gray bars represent divergence values at least one standard deviation above (24.57% − 28.4%) or below (16.79% – 12.90%) the mean. Hatched bars represent divergence values more than two standard deviations above the mean (> 28.46%). Sister genus pairs without standard error bars indicate that only two species were compared to arrive at the average distance between sister genera. Numbers next to genus names indicate the node on Fig 2 corresponding to the genera in that comparison. Asterisks are present to indicate comparisons discussed in detail in the text (lasiurine genera).

(*Euderma* H. Allen, 1892 [76]– *Lasiurus sensu lato*). The average genetic distance among these pairs was 20.68% with a standard deviation of 3.89%. The Shapiro-Wilk test indicated that the null hypothesis of a normal distribution could not be rejected (p > 0.5). The taxon pair *Aeorestes-Lasiurus* was 21.77% ± 1.01% divergent and *Dasypterus* to this pair was 22.91% ± 1.3%, whereas all lasiurines-*Euderma* (labelled as "*Euderma – Lasiurus sensu lato*" in Fig 3) was by far the most divergent sister genus pair at 31.14% ± 1.64.

Note that the finding of Amador et al. [37] of *Euderma* as the sister taxon to lasiurine bats is a novel relationship. Previous studies on vespertilionid relationships (e.g., [77,78]) have lasiurines as sister to *Corynorhinus* H. Allen, 1865 [79]. That sister taxon relationship (*Corynorhinus* to *Lasiurus sensu lato*) results in a 30.0% ± 1.4% difference, still more than any other sister taxon pair in our analysis.

## Within genus analysis

A total of 33 genera (including *Lasiurus sensu lato* and *Eptesicus sensu lato*) with multiple species were examined for interspecific distances (Fig 4; Supplementary S3 Table). Values for these comparisons ranged from 1.84% (*Baeodon*) to 20.26% (*Lasiurus sensu lato*). The average genetic distance among species within the 33 genera was 13.29%, with a standard deviation of 4.11%. Divergence within *Aeorestes* was 14%, *Lasiurus* 15.61%, and *Dasypterus* 16.9%. The vespertilionine genus with the highest value for intrageneric divergence was *Lasiurus sensu lato* with 20.26% average divergence among species.

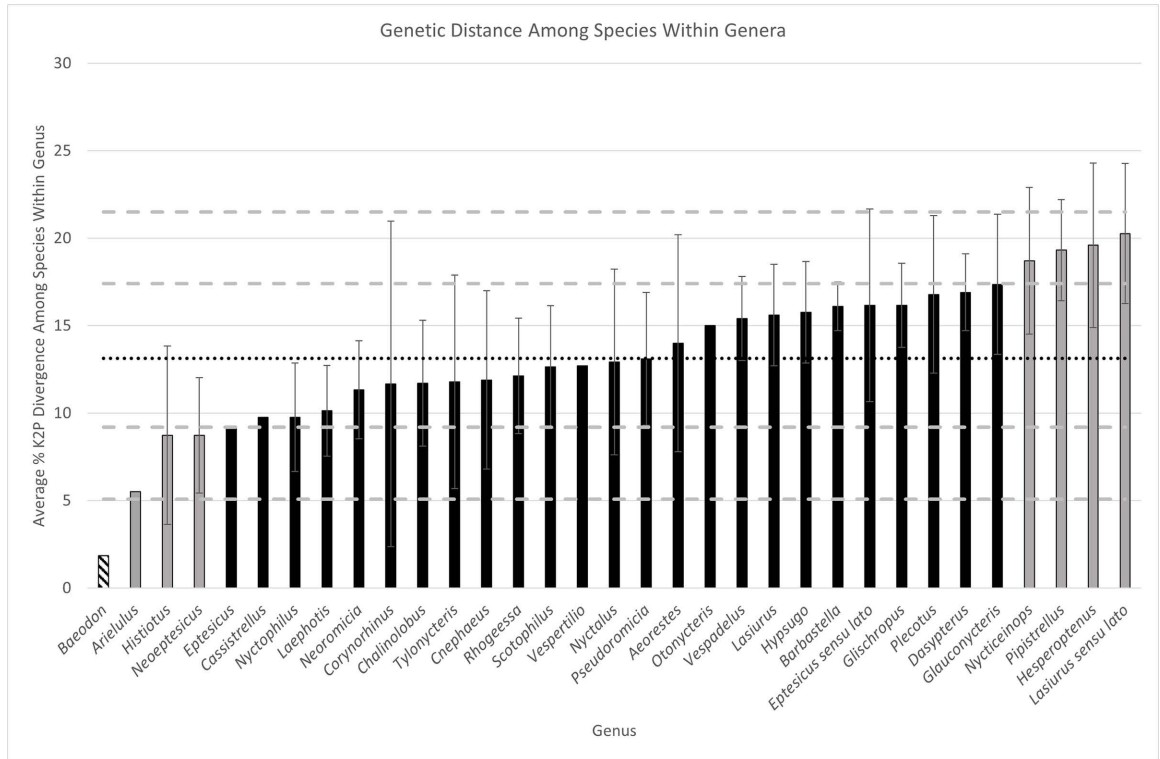

**Fig 4. Genetic divergence among species within polytypic genera of Vespertilioninae.** Mean value (13.13%) is depicted by the horizontal black dotted line. Each standard deviation above and below the mean is depicted by a gray dashed line. Black bars represent divergence values within one standard deviation of the mean (9.19–17.40%). Gray bars represent divergence values at least one standard deviation above (17.40% − 21.50%) or below (5.08% − 9.19%) the mean. Hatched bars represent divergence values more than two standard deviations below (<5.08%) the mean. Within genus distance values that have no standard error bar indicates that only two species were compared for that genus. In this figure, "*Eptesicus*" refers to the arrangement of Cláudio et al. (2023), along with their recognition of *Cnephaeus* and *Neoeptesicus*, also shown here. *Eptesicus sensu lato* refers to the prior arrangement of *Eptesicus* (not including *Histiotus*).

## Case study – Stenodermatinae

For stenodermatine bats, 12 sister genus comparisons were made based on the phylogeny in Supplementary Data S5 Fig., plus *Koopmania – Artibeus*. The average distance among sister genera was 10.97% with a standard deviation of 3.96% (Fig 5; Supplementary Data S4 Table). The Shapiro-Wilk test indicated that the null hypothesis of a normal distribution could not be rejected ($p > 0.5$). The distance from *Koopmania* to *Dermanura* was 11.6% ± 1.0%, whereas *Koopmania* to *Artibeus* was 11.0% ± 1.1%. The range of distances between sister genera in Stenodermatinae was wider than in Vespertilioninae. The minimum distance among sister genera in Stenodermatinae was 3.2% (*Stenoderma* É. Geoffroy Saint-Hilaire [80] to *Phyllops* Peters, 1865 [81]) and the maximum was 16.1% (*Ectophylla* H. Allen, 1892 [76] to a lineage of multiple genera), resulting in a greater standard deviation in stenodermatine bats.

## Discussion

The analysis of genetic distance between sister genera summarized in Fig 3 emphasizes that the taxonomy of Baird et al. [28] recognizing three genera of lasiurines puts those recognized genera (*Aeorestes*, *Lasiurus*, and *Dasypterus*) very close to the average values of divergence between sister genera of vespertilionine bats. The outlier of this analysis is *Lasiurus sensu lato*, exhibiting the highest level of divergence from its sister taxon (*Euderma*). The jump between

*Euderma* – three genera of lasiurine bats (31.14%) and the next highest comparison (*Scotophilus* Leach, 1821 [72] – multiple: 26.33%) is 4.8% different. Nowhere else in the distribution of genetic distances (Fig 3; Supplementary Data S2 Table) is there such a large gap (the next highest jump is 1.69% difference between the "*Rhyneptesicus* – multiple" clade and the "*Arielulus* – multiple" clade). Broken down further, Fig 4 emphasizes why keeping all lasiurine bats together as *Lasiurus sensu lato* is unwarranted in terms of divergence within the genus. *Lasiurus sensu lato* is composed of highly divergent species, more so than any other genus within the subfamily. The next closest in terms of within-genus diversity is *Hesperoptenus*, followed closely by *Pipistrellus* Kaup, 1829 [57]. However, *Pipistrellus* is a problematic genus that likely is paraphyletic and possibly in need of taxonomic revision [82]. Similarly, when *Lasiurus sensu lato* is separated into three genera, those genera still are comprised of diverse species, and that diversity is above the average level of diversity observed within genera of vespertilionine bats (Fig 4).

**TMRCA and genetic divergence**

Our results are congruent with those of Baird et al. [36], who found *Lasiurus sensu lato* to be the most extreme outlier among genera in Vespertilionidae in terms of Time to the Most Recent Common Ancestor (TMRCA). Similarly, Baird et al. [36] found that *Lasiurus*, *Aeorestes*, and *Dasypterus* had approximately average values of TMRCA among vespertilionid genera. The uncertainty surrounding estimates of TMRCA has been used as a rationale for rejecting the recognition

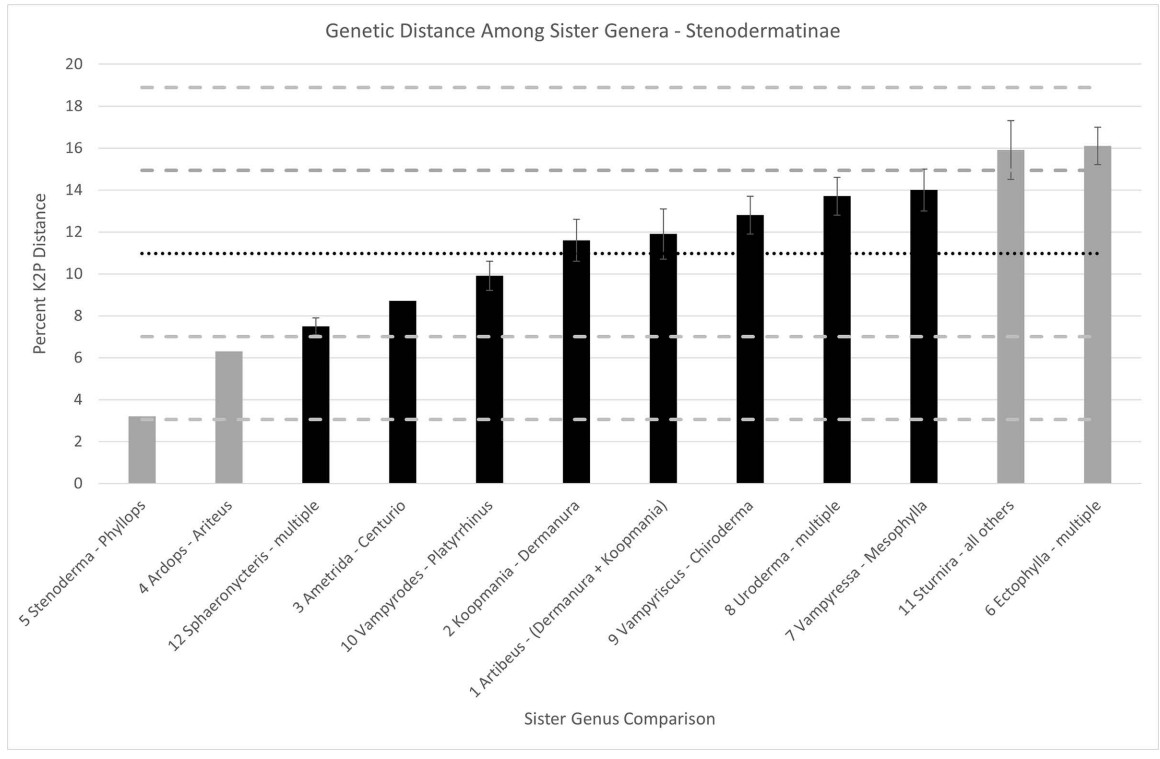

**Fig 5. Genetic distance among sister genera of stenodermatine bats.** Mean value (10.97%) is depicted by the horizontal black dotted line. Each standard deviation above and below the mean is depicted by a gray dashed line. Black bars represent divergence values within one standard deviation of the mean (7.01-14.93%). Gray bars represent divergence values at least one standard deviation above (14.93-18.88%) or below (3.05-7.01%) the mean. Sister genus values that have no standard error bar indicates that only two species were compared. The comparison *Koopmania-Artibeus* is not depicted here, but that value is 11.0±1.1% which is within one standard deviation above the mean. Numbers next to genus names indicate the node in Supplementary S5 Fig. corresponding to the genera in that comparison.

of three genera of lasiurines [34]. However, the high level of variability in TMRCA results from the variability in the fossil age estimates used in the calibration. Genetic divergence, which is the metric upon which the age estimates are based, has no such error or uncertainty in its calculation. It is a metric based on the number of mutations along the branches of the lineages being compared. The number of mutations generally is a function of time, although other factors such as historical effective population size ($N_e$), generation time, incomplete lineage sorting, substitution rate, and selection also can affect branch lengths [83]. This is why sister taxa sometimes have different branch lengths from one another despite having originated from a common ancestor.

The choice of gene to analyze is important because different genes can have markedly different mutation rates: some are more prone to selection or have different modes of inheritance. In this study, we chose to use the mitochondrial *Cytb* gene because of its maternal inheritance, high mutation rate, and small $N_{ef}$. Importantly, it has been the most frequently used genetic marker in evolutionary studies of mammals, and thus more is known about it than probably any other single locus. Its mutation rate closely parallels the 2% per MY of the mtDNA chromosome [84,85]. This same locus was used by Bradley and Baker [51] and Baker and Bradley [52], who showed that most species of mammals differ by about 3% at this locus and that most bat genera have an average of 12% divergence from non-sister species (it is important to note that there is a typographical error in Table 3 of Baker and Bradley [52] – the column labelled "intergeneric" should be "intrageneric," R. D. Bradley, Texas Tech University, in litt.). Our data and the comparisons presented in Baker and Bradley [52] demonstrate that vespertilionid bats are more divergent than other groups of bats, further emphasizing our point that comparisons are most useful when carried out within the next higher rank of taxonomy, rather than compared across distantly related groups. These papers are widely cited as providing validity for recognizing distinct species based upon genetic distance. That hypothesis likewise has been extended here to decisions about generic level status. We therefore hypothesize that comparisons to mean levels of genetic divergence among other closely related genera within the family Vespertilionidae are defensible and likely to be confirmed using other gene loci and analytical approaches to characterize diversity. In response to point 3 of Francis et al. [34], we agree that there is uncertainty with regards to divergence times, especially when the fossil record is poor, as it frequently is for bats. But measures of genetic divergence are much more precise and objective measures are achievable if we choose to develop them within a taxonomic rank. The average genetic divergence in comparisons among 29 pairs of sister genera of Vespertilioninae was 20.68% (Fig 3, Supplementary S2 Table) with the pair *Aeorestes-Lasiurus* falling near the mean, *Dasypterus*-multiple well above the mean, and *Lasiurus sensu lato-Euderma* two standard deviations above the mean. Considering another genetic measure of diversity, average genetic distance among species within a genus, Fig 4 shows the three lasiurine genera to all be above the average of 13.13%. Importantly, it also shows that *Lasiurus sensu lato* is an outlier, having the highest average genetic divergence among the 33 polytypic genera of Vespertilioninae analyzed (20.26%). This finding demonstrates that *Lasiurus sensu lato* consists of highly divergent species more appropriately divided into multiple genera that themselves show above average levels of divergence.

## The question of subgenera

Teta [20] championed the use of the taxonomic category subgenus including for the three genera of lasiurines that we recognize. The author provided a reasoned argument for its use in the recognition of "*diagnosable clades of closely related species*" ( [20]:209). He explained that the use of subgenera provides information about the taxonomic position of different species within the genus. The use of subgenus for the three lineages of lasiurines first was suggested by Ziegler et al. [32] and followed by Novaes et al. [33] and Francis et al. [34].

Alas, there is a fundamental disconnect in Teta [20]—a disconnect that Francis et al. [34] failed to appreciate—between biology and nomenclature. First, as the title of his work implies, Teta ( [20]:209) was discussing "*the advantages of the usage of subgenera as a practical taxonomic rank in mammalian taxonomy*", rather than whether they reflected any underlying biological reality, because (among other reasons) "*subgenera are governed by the rules of the International*

*Commission on Zoological Nomenclature (ICZN)* […]." We argue that many of the taxonomic categories used today and in the past are based in biology not bureaucratic convenience. Certain fundamental tenets of the International Code of Zoological Nomenclature ("*underlying principles*" in the words of the Code) must explicitly be mentioned because they are particularly pertinent to this discussion and are enunciated in the introductory section on principles [21]; namely:

(1) The Code refrains from infringing upon taxonomic judgment, which must not be made subject to regulation or restraint.

(2) Nomenclature does not determine the inclusiveness or exclusiveness of any taxon, nor the rank to be accorded to any assemblage of animals, but, rather, provides the name that is to be used for a taxon whatever taxonomic limits and rank are given to it. [34]

We agree with Teta [20] that the use of subgenera is appropriate for the recognition of diagnosable clades of closely related species, but Fig 3 clearly shows that the species of *Lasiurus* are *not* closely related to species of its sister lineage *Aeorestes* or to the more distantly related *Dasypterus*. They are much more genetically distinct than are species pairs from other well recognized genera such as *Eptesicus* and *Histiotus,* or *Scotomanes* and *Ia* and are an outlier in TMRCA [36]. In response to the second point of Francis et al. [34], we do not agree that the use of the category subgenus is appropriate when levels of genetic diversity within and between the named lineages are as diverse as other genera within the group's family or subfamily.

## Nomenclatural stability

The four papers criticizing the recognition by Baird et al. [28,35,36] of three genera of lasiurine bats all cited the ICZN's [21] stated principle of nomenclatural stability as a primary reason to classify these taxa as subgenera rather than genera. It should be noted that the Code does not define nomenclatural stability, nor does it suggest that taxonomic decisions should be made for reasons such as the ease of finding relevant literature in a computer literature search. If such searches are done properly, all name usages will be easily and rapidly identified. We must recognize instead that nomenclatural stability does not mean the absence of change, but rather consensus of treatment, and consensus of treatment will be achieved when the underlying biological phenomena and relationships are revealed, rather than obscuring them. The recommendation of Francis et al. [34] that the three lineages of lasiurines be considered as subgenera will not provide nomenclatural stability. On the contrary, it will assure the continued state of taxonomic instability that long has characterized this group of bats (e.g., see Fig 2 in [36]:286) because it is not anchored to biological differences. In fact, neither Francis et al. [34], nor any of the other opinion papers provided any new biological data or analyses to demonstrate that these lineages should be considered subgenera. They concluded this merely to avoid having to change the scientific names of the genera *Dasypterus* and *Aeorestes* and species contained therein. This will only delay arriving at a more accurate taxonomy because future studies of lasiurines will continue to show the high levels of genetic diversity within and among these three lineages, and that this diversity is comparable to other vespertilionine genera, not subgenera. Such inaccuracies in taxonomy subvert efforts to understand patterns of evolution, adaptation, and biogeography that often are based on genus-level taxonomy as argued previously by Helgen et al. [10].

Furthermore, there is no mention of the second principle of the ICZN: that nomenclature does not determine the inclusiveness or exclusiveness of any taxon nor indeed its rank. Rather, nomenclature provides the name, to be used whatever rank is given to it. This topic was addressed in a passage from McKenna ( [86]:27):

"*To the critics who would ask, "Where will all this proliferation of names and ranks end?," I suggest that if the terms are not found useful to convey exact genealogical meanings dictated by phylogeny, then they can be ignored by those who so choose. They are, however, based upon cladistic principles, not on "art" or caprice.*"

McKenna [86] goes on to add that he considered several ways to avoid the problem of proliferation of ranks. Among them, "*by taking Hennig at his word and indicating rank by a number representing millions of years, inasmuch as rank is proportional to relative recency of common ancestry in cladistic systems*" ( [86]:27). McKenna eventually discounted that

method, noting that such a "*scheme requires knowledge of the time of actual branching, which in fact is never known but only estimated, in many cases not even approximately*." ( [86]:27). The increasing number of fossils has vastly improved our knowledge of the fossil record, and the potential age of many groups; this is not limited to mammals. One might argue that those advances in paleontology have been matched with advances in molecular biology: sequencing has progressed past a partial mitochondrial gene, through single mitochondrial genes, and even past multiple nuclear and mitochondrial genes to whole genomes [53]. These advances have come hand in hand with advances in analytical software, such that we now are able to estimate the time of branching as accurately as the fossils allow us to do so.

## Morphological distinction

Historically, morphological characters have been used almost exclusively in decisions regarding generic level taxonomy. With regards to lasiurine bats, the taxonomic arrangement of a single genus within the tribe was established by Handley [87]. Prior to this, there were two genera, *Lasiurus* (red and hoary bats) and *Dasypterus* (yellow bats). His Table 3 (p. 475) lists "*some of the conspicuous differences between species of* Lasiurus *and* Dasypterus." The quote is from p. 473. In fact, his Table 3 is a list of 12 morphological characters that differentiate the three species groups, considered here as genera. On p. 473 he listed seven morphological and reproductive characters (or more as some could be considered multiple characters) that he described as follows: "*More impressive are the following similarities linking these nominal genera and distinguishing them from other vespertilionids and in some cases from all other bats*". Indeed, some of these characters are impressive, such as the presence of four mammae and litter sizes of two or three pups (almost all bats have a single young). But his list described characters that define the tribe Lasiurini.

Nonetheless, the 12 morphological characters in his Table 3 that differentiate the species groups also can be considered significant and may be used just as effectively to define three genera. Handley's [87] entire discussion of the taxonomic implications of his morphological data was given in two sentences (p. 473): "*It seems more reasonable to stress the important similarities of these bats and regard them as congeneric, rather than to stress the insignificant differences and regard them as representing distinct genera. I do not believe that* Dasypterus *is useful even as a subgenus*." It is remarkable that this clearly subjective conclusion is the basis of the taxonomy of lasiurine bats that our critics so forcefully defend. What makes the characters in Handley's Table 3 insignificant in the opinions of Handley and, apparently, these authors? There is no definition given of what differentiates a significant from an insignificant character in any of the papers discussing lasiurine taxonomy and no scale exists to rate morphological characters as being indicative of subgeneric or generic rank. However, as we show here, genetic divergence distinctively and objectively shows that the three species groups of Handley are divergent at a level comparable to other genera of vespertilionines. Handley's [87] conclusion that the three lineages of lasiurines comprise a single genus therefore is based on faulty reasoning. The high degree of morphological divergence that distinguishes Lasiurini from other vespertilionid tribes should in no way influence how the lineages contained within the tribe are classified.

Novaes et al. [33] addressed the issue of the morphological distinction of the three lineages of lasiurines. Remarkably, in support of using morphology they state that genetic distance and divergence times are not evidence for the recognition of genera because these measures are not comparable among taxa. The only example they gave is from primates, where *Tarsius* (Tarsiidae) is ca. 45 MY old [88] and *Pan* and *Homo* (Hominidae) diverged ca. 6 MYA [88]. This hardly seems an apt comparison to bats, and Novaes et al. [33] failed to make any attempt to show that morphology is scalable or comparable among taxa. With regards to the generic status of yellow bats (*Dasypterus*), Novaes et al. ( [33]:439) said: "*there are no clear phenotypic discontinuities supporting Tate's [29] hypothesis, which has been subsequently refuted (see [77,78])."* This is stated even though they previously discussed the key dental difference in which *Dasypterus* has one less pair of premolars that either is always present in species of *Lasiurus* or variable in number in species of *Aeorestes*. They did not list any of the other morphological differences given in Handley [87], such as size (forearm length), relative length of rostrum, sagittal crest weak or strong, and height of coronoid process. Furthermore, neither of the papers cited by Novaes

et al. [33] refuted Tate [29]; rather, Hoofer and Van Den Bussche [77] and Roehrs et al. [78] make the same point, i.e., that their genetic studies do not resolve the position of the hoary bat relative to red bats. In fact, Hoofer and Van Den Bussche ( [77], p. 26) stated the following: *"In contrast, mtDNA analysis demonstrates marked separation between yellow and red bats (and hoary bats), but this may not warrant generic revision because the position of hoary bats is unresolved. Previous recognition of Dasypterus was based primarily on support for sister relationship between red and hoary bats, a relationship clearly unresolved in this study (Fig. 3)."* Considering this statement, we conclude that because Baird et al. [28,35] clearly resolved the hoary bat and red bat lineages to be sister, it is consistent with the findings of Hoofer and Van Den Bussche [77] to conclude that generic status is warranted for *Dasypterus*.

**2 Sigma Genus Concept**

There is no precisely defined operational definition of genus that is generally recognized in the mammalian systematics literature. In fact, mammalian systematists consider a broad range of morphological and genetic characters and other forms of evidence to be relevant to a generic revision. Cláudio et al. ( [55]:10) suggested the following: *"Taxonomic decisions at the generic level should consider phylogenetic relationships, synapomorphic character status, phenotypic distinctiveness, and ecological factors in the formal recognition of distinctiveness among clades [*89,90*]. Also, the zoological nomenclature "should convey evolutionary relationships, diversity, divergence, and the potential to clarify conservation priorities"* ( [36]: 285). Other factors that have been considered as indicative of generic level distinctions include biogeography, species distributional overlap, divergence times [34], and the inability to form intergeneric hybrids [8].

The genus concept was first established in the 10th edition of the *Systema Naturae* by Linnaeus in 1758 [1]. Thus, our existing taxonomy is built upon almost 300 years of established scholarship arriving at a consensus concept (absent a definition) of genus. Even though genus is undefined in the mammalogical literature, the existing genera are generally well established and accepted, despite such examples of subdivision as *Lasiurus*, *Eptesicus, Artibeus* and *Tamias* discussed in this paper. As such, evidence of substantial and significant departures in the degree of differentiation from typical intergeneric genetic distances (as is the case of Lasiurini) should be construed as a strong indication of a need for taxonomic revision.

Here, we provide a genetic-based operational definition of the 2 Sigma Genus Concept for use in mammalogy as follows: *A genus is a formally recognized taxon containing one species or a monophyletic lineage of species. Genera are morphologically and/or genetically distinct from other genera, and genetic as well as morphological and other heritable biological features can be used for diagnoses. Genetic divergence estimates between sister genera should comprise a range between those of subgenera or congeneric species and higher taxa such as tribe, subfamily, or family. Operationally using genetic distance values, this range is set at two standard deviations below the mean as the minimum value for a valid genus, and two standard deviations above the mean as the minimum value for a new higher taxon such as tribe or subfamily.*

We used, and recommend, the following 5 steps to determine the status of a genus. 1) Determine that all genera or lineages proposed as genera are monophyletic. 2) Refer to the most complete phylogeny of the inclusive higher category to determine all sister-genus comparisons. 3) Calculate sister-genus genetic distances for all genera. 4) Assuming a normal probability distribution, determine the mean and standard deviation of the calculated genetic distances. 5) A proposed new genus whose sister-genus genetic distance falls below 2 standard deviations of the mean does not qualify for generic status.

Considering these steps, our method is novel in that it uses the statistical distribution of sister divergences within a higher category to guide the allocation of generic status to monophyletic lineages. Moreover, this novel method is phylogenetically based; sister genera are identified based on the best phylogeny available for the higher category. Considering this definition, genetic divergence between a newly proposed genus and its sister genus or sister lineage should fall within the range of well accepted sister-genus/lineage pairs within the higher taxon that contains the proposed new genus. The decision to recognize a new genus is well supported if it falls within one standard deviation of the mean value of genetic

distance comparisons between sister genera for the higher category. If it falls below one standard deviation but less than two, generic status should still be considered an acceptable decision. But the question of generic status is not as secure and other genetic measures might not support it. If it falls below two standard deviations the proposed new genus should not be accepted. Likewise, sister genus genetic distances beyond two standard deviations above the mean can be considered as support for the recognition of a higher category such as tribe.

Changes to a genus-level taxonomy should be made when a new genus is discovered, when an existing genus is found to be paraphyletic, when intergeneric distances are too low or too high, or when an existing genus is found to contain unnamed lineages with divergence levels that fall within the range of genera. Taxonomic revisions of any taxon, not just mammals, can make use of the 2 Sigma Genus Concept provided there is adequate coverage of genetic diversity for the potential generic-level lineages. Thus the validity of both newly proposed, as well as currently recognized genera, can be tested. Following these principles, making modifications to generic-level taxonomy will result in a classification that better reflects the biology of the higher taxon and ultimately contribute to taxonomic stability of the revised clade. Note that we are not saying that there should be a particular range of genetic divergence that should denote generic level distinction among all forms of life, or among all vertebrates, or even among all mammals. Rather, we are saying that the nearly 300 years of progress in taxonomy has provided a baseline nomenclature from which can be extracted an acceptable range of genetic diversity upon which to establish generic level taxonomy for a higher taxon. As changes are made to generic-level taxonomy within a family or subfamily, that higher level group will have genera that are thereby closer to the mean value and will not have genera >2 standard deviations from the mean (either by combining genera with very low divergence or splitting genera with very high divergence). This now is the case in vespertilionines following the taxonomy of Baird et al. [28].

Thus, we propose a comparative process that makes use of existing taxonomies based on the most recent revisions as systematic hypotheses. Previous proposals that have tied taxonomic levels to divergence times, such as Avise and Johns [9], are similar to ours in that both proposals are based on genetic divergence. But the added step of calculating mutation rates based on fossil dating estimates would seem to be an unnecessary complication. Moreover, while the goal of a universal standardized taxonomic ranking based on divergence time, and hence indirectly on genetic divergence, is laudable, we do not believe that it is a more laudatory goal than recognizing the current taxonomy as appropriate and worthy of being sustained. While the genetic divergence estimates of genera of vespertilionids are likely to be similar to those of shrews and other mammals, there is no reason to believe that they will be similar to tardigrades or flies, nor should they be because the biology of such diverse animals is so dissimilar.

## Case study, stenodermatine bats

We used phyllostomid bats in the subfamily Stenodermatinae as a test of the 2 Sigma Genus Concept and to compare sister genus distances in a more distantly related group of mammals. The Stenodermatinae includes two tribes, Stenodermatini and Sturnirini. At issue in Stenodermatini is the taxonomic status of the monophyletic clade containing the genera *Artibeus*, *Dermanura,* and *Koopmania* (Supplementary Data S5 Fig.). Some authors treat *Dermanura* and *Koopmania* as subgenera of *Artibeus*, whereas others recognize one or both genera as distinct from *Artibeus*. The Mammal Diversity Database [38] and batnames.org [39] both recognize *Dermanura* as a genus but treat *Koopmania* as a synonym of *Artibeus*. The debate over genera vs. subgenera in *Artibeus* therefore is a similar situation to lasiurine bats in that the taxonomic changes were not initiated by the discovery of paraphyly.

The mean value for distance among sister genera in Stenodermatinae is 10.97% with standard deviation of 3.96% (Fig 5), whereas these values in Vespertilioninae are 20.68% (mean) and 3.89% (SD; Fig 3). The distances between *Koopmania* and *Dermanura* and *Koopmania* to *Artibeus* fall within one standard deviation of the mean (Fig 5); recognition of these genera therefore is well-supported based on the 2 Sigma Genus Concept. In contrast, some intergeneric relationships in this subfamily are more questionable as to the validity of generic status. For example, among the white-shouldered bats (denoted by nodes 4 and 5 in Supplementary Data S5 Fig.) *Stenoderma* and *Phyllops* are only 3.2% different and barely within the two

standard deviation threshold (Fig 5). With such a low distance value they barely qualify as distinct species in other groups of chiropterans, let alone distinct genera. Yet, they are recognized as genera both by Mammal Diversity Database [38] and batnames.org [39] as are the five other monotypic genera of white-shouldered bats (*Sphaeronycteris, Ardops, Ariteus, Stenoderma* and *Phyllops*). The most divergent pair of genera in this group are *Ametrida* and *Centurio* at 8.4%, which is still lower than any other pair of genera not included in the white-shouldered group. Thus, the white-shouldered bats stand out as a group that appears to have experienced rapid morphological change which is captured in the 2 Sigma Genus Concept.

The above example also highlights the importance of context when deciding generic status. The intergeneric distances among stenodermatine bats are quite different from those of vespertilionine bats, indicating that sufficient morphological changes to recognize genera have happened more recently in stenodermatines than in vespertilionines. In deciding the generic status of a group of organisms, the comparison always should be made in the context of the next higher taxonomic group. We do not intend our range of intergeneric distances among vespertilionines to be used to make generic-level decisions in other groups of organisms. Within vespertilionines, however, these values are quite useful to examine the validity of new genera, as in the newly erected genera *Afronycteris* and *Pseudoromicia* [58]. Monadjem et al. [58] report the distances of these new genera to other recognized vespertilionine genera to be 15.7% − 19.6% [58], which would be divergent enough to be supported as genera by the data we have presented here.

The 2 Sigma Genus Concept has been examined with two test cases (vespertilionine and stenodermatine bats) and the methods are robust in those cases. Although these two major groups of bats are well represented in molecular studies, other less well studied groups of mammals, or non-mammalian taxa, might not lend themselves well to this approach at the present time. However, there is hope even for the most least well studied groups that in the near future genetic and even genomic data will be available. As an example, the Earth Biogenome project proposes to sequence the genomes of all living species on the planet in ten years (www.earthbiogenome.org). Whereas for hundreds of years the world's museums have been filled with the preserved morphological remains of mammals and other taxa, a time will come when comprehensive genetics and genomics data will be at the fingertips of scientists and students everywhere in the world. This democratization of taxonomy will revolutionize the field and methods such as the 2 Sigma Genus Concept presented here will find broad appeal. In the meantime, the concept can be further tested with other taxa and modeled to examine how factors such as tree shape, or in cases when sister genus genetic distances are not normally distributed, affect the concept. Further, the model is not exclusive to *Cytb* data. It should also be tested using other genes or genomes.

## Conclusions

We define a genus by the following criteria:

1. A monophyletic lineage that is morphologically and/or genetically distinct from other such lineages.

2. The level of genetic differentiation between a new genus and its sister genus or lineage lies within two standard deviations of the mean divergence levels of other sister genus pairs within the same higher-level taxonomic category. Beyond two standard deviations below the mean indicates generic status is not warranted and beyond two standard deviations above the mean suggests a higher category such as tribe or subfamily.

3. Genetic-based characters such as morphology or behavior are useful as diagnostic characters but are not scalable and thus not useful in determining generic status.

The recognition of three genera within Lasiurini meets the criteria of the 2 Sigma Genus Concept above.

1. Each genus (*Aeorestes, Lasiurus, Dasypterus*) constitutes a monophyletic group (Fig 2).

2. Intergeneric divergence levels between *Lasiurus* and *Aeorestes* (21.77%), and between *Dasypterus* and *Aeorestes + Lasiurus* (22.91%), are within one standard deviation of the mean value for Vespertilioninae (mean 20.68%; Figs 3 and 4).

3. Unique morphological characters are diagnostic for each genus (see key to genera below).

Maintaining *Lasiurus sensu lato* would violate the 2 Sigma Genus Concept because the level of genetic divergence among species within *Lasiurus sensu lato*, and between *Lasiurus sensu lato* and its sister genus (*Euderma*) shows *Lasiurus sensu lato* to be an extreme outlier (greater than 2 standard deviations above the mean) among vespertilionine bats (Figs 3 and 4). Criterion 2 therefore is violated. We conclude that the clade is consistent with being restricted to a higher-level taxonomy (tribe Lasiurini) and subdivided into genera that are close to the average level of intergeneric divergence among vespertilionine bats and thus clearly deserve generic status.

The taxonomic arrangement of Baird et al. [28] has been accepted by much of the mammalian taxonomic community. The validity of this arrangement was extensively discussed in a recent paper by Mora and Ruedas [49]. Among their conclusions on this topic are the following:

1. The application of *Dasypterus* for yellow bats has been extensively used in the past and *"therefore follows long established norms and practice"* ( [49]:457).

2. Many authors are following the suggested taxonomy of Baird et al. [28], including the use of *Aeorestes* for hoary bats. A literature search from early 2023 resulted in 99 occurrences for *Aeorestes villosissimus*, versus two results for *Lasiurus villosissimus* ( [49]:458).

3. On the adoption of *Aeorestes*, *"we instead consider this settled precedent…"* ( [49]:458).

### Key to the genera of lasiurine bats

As noted by Handley [87], several characters are synapomorphic among all species of lasiurine bats. These characters include: four mammae and average of 2–3 young per litter; spiral effect in scale arrangement on hairs; reduction of sebaceous glandular tissue and location of the submaxillary salivary gland in the facial area; bright coloration; baculum short, J-shaped, with high base and narrow shaft; distally enlarged and spiny penis; furry interfemoral membrane. These characters, therefore, differentiate tribe Lasiurini.

### Genera within Lasiurini can be identified as follows

1a.  Yellow fur; variable in size; $P^1$ always absent; sagittal crest strong….*Dasypterus*

1b.  Reddish or multi-colored fur ranging from dark brown to gray with a whitish tip; $P^1$ present or variable; sagittal crest weak…2

2a.  Larger size (forearm 46–57 mm) with hoary-gray pelage or if red with no contrast between dorsal and ventral coloration; pale orange or whitish mottled wings on both sides of the forearms, metacarpals, and digits….*Aeorestes*

2b.  Smaller size (forearm 37–49 mm) with fur coloration reddish and contrast between dorsal and ventral pelage that is frosted…*Lasiurus*

### Supporting information

**S1 Table. List of species examined.** Included in this table are the GenBank accession numbers for *Cytb* sequences used in the sister genus analysis and within genus analysis. If a genus or species within Vespertilioninae does not appear on this list, it is because that taxon was not represented by *Cytb* sequences on GenBank. Taxa included in the case study of Stenodermatinae are also listed.
(XLSX)

**S2 Table. Genetic distances (K2P) among vespertilionine sister genera comparisons for *Cytb*.**
(XLSX)

**S3 Table. Genetic distances (K2P) within genera of Vespertilioninae.**
(XLSX)

**S4 Table. Genetic distances (K2P) among stenodermatine sister genera comparisons for *Cytb*.** The comparison *Koopmania – Artibeus* is shown below but not used in the calculation of mean and standard deviation (because *Koopmania – Dermanura* was used in this calculation; the sister taxon to *Koopmania* is unresolved – see discussion in text). Numbers correspond to nodes in Supplementary S5 Fig.
(XLSX)

**S5 Fig. Relationships among stenodermatine bats, derived from Garbino and Tavares [67], with *Koopmania* added.** See text for discussion on relationships among *Artibeus*, *Dermanura*, and *Koopmania*. Numbers above nodes correspond to sister genera relationships analyzed. Distances among these sister genera are given in Supplementary S2 Table.
(TIF)

## Acknowledgments

We thank D. J. Schmidly, R. D. Bradley, J. I. Baird, and J. A. DeWoody and his lab members for discussion and many helpful comments on an early draft of the manuscript. J. I. Baird suggested the term "2 Sigma Genus Concept".

## Author contributions

**Conceptualization:** Amy B. Baird, Janet K. Braun, Mark D. Engstrom, Burton K. Lim, Michael A. Mares, Luis A. Ruedas, John C. Patton, John W. Bickham.

**Data curation:** Amy B. Baird.

**Formal analysis:** Amy B. Baird, Luis A. Ruedas, John W. Bickham.

**Investigation:** Amy B. Baird, John W. Bickham.

**Methodology:** Amy B. Baird, John W. Bickham.

**Writing – original draft:** Amy B. Baird, John W. Bickham.

**Writing – review & editing:** Amy B. Baird, Janet K. Braun, Mark D. Engstrom, Burton K. Lim, Michael A. Mares, Luis A. Ruedas, John C. Patton, John W. Bickham.

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
