## [Decision Letter · Decision Letter 0]

Dear Dr. Baird,

Thank you for submitting your manuscript to PLOS ONE. After careful consideration, we feel that it has merit but does not fully meet PLOS ONE’s publication criteria as it currently stands. Therefore, we invite you to submit a revised version of the manuscript that addresses the points raised during the review process.

We look forward to receiving your revised manuscript.

Kind regards,

Pablo Colunga-Salas

Academic Editor

PLOS ONE

3. We note that Figure 1 in your submission contain copyrighted images. All PLOS content is published under the Creative Commons Attribution License (CC BY 4.0), which means that the manuscript, images, and Supporting Information files will be freely available online, and any third party is permitted to access, download, copy, distribute, and use these materials in any way, even commercially, with proper attribution. For more information, see our copyright guidelines: http://journals.plos.org/plosone/s/licenses-and-copyright.

Additional Editor Comments:

Dear authors,

Based on the comments of two reviewers, I consider that you should address the major comments on your manuscript. Although it is a work that has great potential to be published and contains relevant information on the subject, the number of individuals used within the current genus *Lasiurus* to divide it into the three proposed genera is an issue that you should address.

I trust that you will be able to address the comments requested. Please, if you require more time to make the necessary changes, do not hesitate to write to me.

Sincerely,

Reviewers' comments:

Reviewer's Responses to Questions

**Comments to the Author**

1. Is the manuscript technically sound, and do the data support the conclusions?

Reviewer #1: Partly

Reviewer #2: Yes

2. Has the statistical analysis been performed appropriately and rigorously?

Reviewer #1: Yes

Reviewer #2: Yes

3. Have the authors made all data underlying the findings in their manuscript fully available?

Reviewer #1: Yes

Reviewer #2: Yes

4. Is the manuscript presented in an intelligible fashion and written in standard English?

Reviewer #1: Yes

Reviewer #2: Yes

Reviewer #1: This manuscript is a significant effort to resolve the taxonomic status of the names associated with Lasiurus (Lasiurus, Dasypterus, and Aeorestes). While the proposed operational criterion makes sense, I'm not convinced that such a low taxonomic sampling within each genus or subgenus is adequate. Using only one species to represent the divergence of a genus cannot sufficiently justify this kind of taxonomic decision. The lack of complete gene sequences for all species does not warrant reducing the data to just one species. It is challenging to define a minimum number of species, but I would suggest including at least 30% of the recognized species for adequate taxonomic sampling.

Additionally, authors should clarify whether the proposed operational criterion is intended solely for newly proposed genus-level names or if it should also be applied to assess the taxonomic distinction of existing names in other groups of organisms (bats and beyond). Similar to other previously used criteria, I view it as an indicator for further review before making taxonomic decisions, rather than a single definitive criterion.

Other comments are directly included in the attached PDF.

Reviewer #2: Comments to the Authors

The study addresses a highly interesting topic in mammalian systematics and taxonomy: the definition and delimitation of taxonomic groups at the generic level. To achieve this, the authors focus on the bat genus Lasiurus, a taxon whose classification remains controversial due to differing opinions on whether it should be divided into three subgenera or three separate genera, depending on the consulted literature. In my opinion, the most significant contribution of this study lies in the proposal of a new genus concept based on genetic distance comparisons without establishing a strict threshold (as is commonly done at the species level). This approach is practical, considering that genetic divergence varies across groups/lineages depending on their evolutionary history and other factors.

General Comments:

The manuscript effectively contextualizes the controversy by referencing other authors who prefer to maintain Lasiurus as a single genus with subgenera. The explanation of why taxonomic stability should not override biologically meaningful nomenclature is well-appreciated. However, the proposed genus concept presents some methodological limitations that should be considered:

1) Use of a single gene (Cytb): While the authors acknowledge this limitation and justify their choice by noting that Cytb is the most commonly used gene in mammalian studies, future research would benefit from comparisons with nuclear or genomic data whenever available.

2) Taxonomic representation: The study includes a good representation of genera (and DNA sequences) from the subfamilies Vespertilioninae and Stenodermatinae. However, robust phylogenetic data are not always available for all mammalian groups, which poses a limitation for the broader applicability of the proposed genus concept. This issue should be discussed in greater depth in the Discussion section.

3) The “2 Sigma” criterion: This statistical approach is sensitive to the normality of the distribution and to the representativeness of each pair of sister genera. What would happen in cases that do not conform to normality (due to evolutionary history, recent radiations, or very ancient divergences)? It would be useful to include a discussion on such scenarios.

4) Morphological differences and their role in the concept: The criterion definition of “The 2 Sigma Genus Concept” implicitly considers morphological differences, and a key for recognizing the three genera within Lausurini is proposed. However, this study does not present morphometric analyses, which could have further supported the genetic criteria. Future studies should include more comprehensive morphometric analyses (e.g., skull geometric morphometrics) to strengthen the validity of the proposed genera.

Particular Comments:

Introduction:

Lines 66-68: I suggest to support this statement with cites.

Lines 134-135: It is suggested to include a size scale in Figure 1. If available, incorporating skull images of the species would be highly beneficial.

Materials and Methods:

The information in this section appears somewhat repetitive and lacks clarity in some parts. Since the genus concept is based on mean values and standard deviations, a more detailed explanation of this methodology is suggested (even if it seems statistically simple and evident). Additionally, it is recommended to divide the Materials and Methods section into subsections for better readability (this suggestion also applies to Results).

Lines 202-203: Should be cited as Amador et al. (24; their supplementary Fig. S6).

Lines 206-210: Due to its large size, Table 1 might be better placed as an Appendix.

Line 235: Supplementary Data S4 does not exist. This should be corrected in the supplementary materials. The phylogenetic tree of Stenodermatinae is labeled as “S1 Fig Stenodermatinae tree.”

Lines 240-241: Which software was used for normality tests? This information should be included.

Results:

Line 254: Why was only one representative per genus used? How does this choice influence potential biases in the analysis? To properly compare polytypic genera, all available species should be included to represent the genetic variability of the genus.

Line 257: The evolutionary model (K2P) should be included on the y-axis of Figure 3.

Lines 264-267: Care should be taken to avoid writing in a discussion-like tone in the Results section.

Line 280: Should read Cláudio et al. (42). Additionally, the accent on Cláudio should be corrected throughout the manuscript.

Lines 283-285: It is suggested that Supplementary Material S3 should contain the phylogeny of Stenodermatinae, while Supplementary Material S4 should include the results of comparisons between sister genera in this subfamily. Furthermore, intergeneric comparisons in Stenodermatinae should be presented as a figure in the main text (as done for Vespertilioninae), since these results need to be well-represented and should not be relegated to supplementary material.

Discussion:

Lines 317-321: These statements should be supported by citations.

Line 354: “He” should be changed to “The autor”.

Lines 374-379: In my opinión, this paragraph appears overly subjective and polemical, which is not appropriate for a scientific paper. It is recommended to rephrase it in a more neutral tone while still addressing the key concerns.

Line 417: Should be McKenna (55) instead of just the author's last name.

Line 656: Should be Supplementary Material S3.

Line 662: There are typographical errors and missing information in some references, such as 11, 21, 27, and 48.

**Do you want your identity to be public for this peer review?** For information about this choice, including consent withdrawal, please see our Privacy Policy

Reviewer #1: No

Reviewer #2: No

---

## [Author Response · Author response to Decision Letter 1]

7 Apr 2025

Dear all,

Thank you for the helpful comments. We have included detailed responses to each comment. These can be found in the Cover Letter.

Sincerely,

Amy Baird

---

## [Decision Letter · Decision Letter 1]

The 2 Sigma Genus Concept in Mammalogy: Lessons from Lasiurus

PONE-D-25-05977R1

Dear Dr. Baird,

We’re pleased to inform you that your manuscript has been judged scientifically suitable for publication and will be formally accepted for publication once it meets all outstanding technical requirements.

Kind regards,

Pablo Colunga-Salas

Academic Editor

PLOS ONE

Additional Editor Comments (optional):

Dear authors,

On behalf of both reviewers, I thank you in advance for your effort in addressing the comments, which I believe helped enrich your work. I know this contribution will be of great interest to mammalogists, especially those specializing in bat taxonomy.

All the best

Reviewers' comments:

Reviewer's Responses to Questions

**Comments to the Author**

Reviewer #1: (No Response)

Reviewer #2: All comments have been addressed

2. Is the manuscript technically sound, and do the data support the conclusions?

Reviewer #1: Yes

Reviewer #2: Yes

3. Has the statistical analysis been performed appropriately and rigorously?

Reviewer #1: Yes

Reviewer #2: Yes

4. Have the authors made all data underlying the findings in their manuscript fully available?

Reviewer #1: Yes

Reviewer #2: Yes

5. Is the manuscript presented in an intelligible fashion and written in standard English?

Reviewer #1: Yes

Reviewer #2: Yes

Reviewer #1: I couldn't find the cover letter addressing the questions raised in the previous draft. However, the revised draft includes specific modifications aimed to solve those comments, and I appreciate the work by the authors in responding our observations.

Reviewer #2: Dear Authors,

Thank you for addressing most of the comments and suggestions I provided during the previous round of review. Best of luck with your manuscript.

**Do you want your identity to be public for this peer review?** For information about this choice, including consent withdrawal, please see our Privacy Policy

Reviewer #1: No

Reviewer #2: No

---

## [Editor Report · Acceptance letter]

PONE-D-25-05977R1

PLOS ONE

Dear Dr. Baird,

I'm pleased to inform you that your manuscript has been deemed suitable for publication in PLOS ONE. Congratulations! Your manuscript is now being handed over to our production team.

Kind regards,

on behalf of

Pablo Colunga-Salas

Academic Editor

PLOS ONE